# PointVLM: Multi-Modal Vision-Language Model for CAD Model Understanding via Point Cloud Integration

## Abstract

In computer-aided design (CAD) and engineering, understanding complex CAD models remains a critical challenge. Existing methods struggle with integrating geometric features due to the lack of 3D modality and the difficulty of modal fusion. To address this, we introduce PointVLM, a novel multi-modal vision-language model that bridges 3D point cloud processing with vision and natural language understanding to enable precise CAD model interpretation. PointVLM leverages a 3D encoder to grasp 3D features from the point cloud of the object in addition to vision and language modalities. By combining Qwen2.5-VL architecture, PointVLM fuses three kinds of modality features using a learnable projector module, enabling context-aware interactions between geometric and semantic properties. We further build a pipeline which takes CAD file and instruction as input, automatically samples point clouds and renders multi-view images, and finally outputs responses. Experiments show that PointVLM outperforms existing methods on both generative 3D object classification and 3D object captioning tasks. The source code and pre-trained models will be available at `MASKED_URL`.

## 1 Introduction

Computer-aided design (CAD) has fundamentally transformed engineering and manufacturing by enabling precise digital representations of physical objects through parametric modeling and geometric optimization. However, as CAD models evolve into intricate multi-object systems with hierarchical assembly relationships, interpreting complex CAD models remains a persistent challenge due to the intrinsic complexity of 3D spatial reasoning and topological coherence.

Traditional approaches to CAD interpretation rely on human expertise for geometric analysis, material property mapping and assembly validation. While recent advances in large language models (LLMs) like DeepSeek-V3 (Liu et al., 2024a) have revolutionized textual reasoning, their inherent sequential processing architecture fundamentally misaligns with the non-sequential nature of 3D representations.

Emerging vision-language models (VLMs) such as Qwen2.5-VL (Bai et al., 2025) have demonstrated promising capabilities in cross-modal reasoning by aligning visual features with textual descriptors, but the application to CAD model understanding remains nascent. Existing methods such as Liu et al. (2024b) attempt to fuse LLMs with 2D images enable 3D comprehension but struggle with problems such as depth ambiguity, occlusion and viewpoint dependency. This gap highlights two fundamental challenges: 1) The absence of specialized 3D spatial reasoning mechanisms that can handle unordered point clouds while preserving topological relationships, and 2) The lack of alignment between geometric primitives and linguistic descriptors in multi-modal fusion architecture.

To address these limitations, we propose PointVLM, a novel multi-modal architecture that synergizes PointBERT (Yu et al., 2022) for 3D geometric encoding with Qwen2.5-VL's (Bai et al., 2025) multimodal reasoning capabilities. PointVLM leverages a PointBERT-based point encoder to learn representations of point cloud. To align point cloud, image and text features in the same space, we propose a pre-training framework based on ULIP (Xue et al., 2023) in the pre-training stage. PointVLM also adapts state-of-the-art visual language models to process 3D spatial relationships

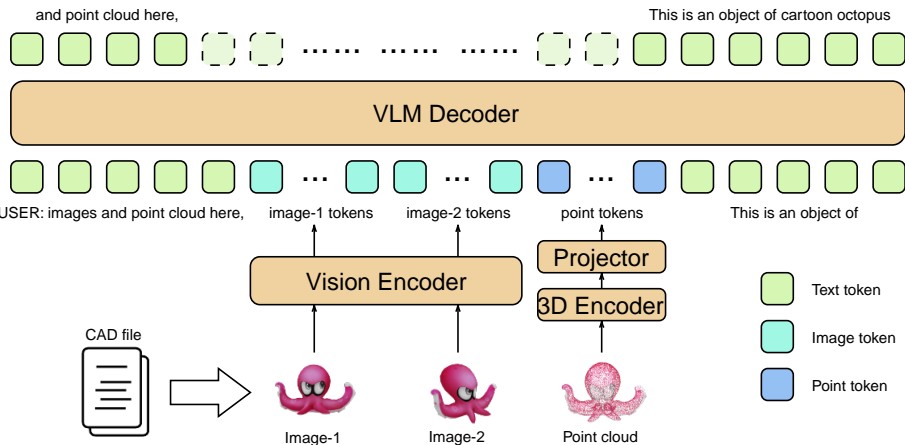

Figure 1: An overview of PointVLM pipeline. After getting CAD file and user prompt inputs, multi-view images are firstly rendered, and point cloud is sampled at the same time. Then using vision and 3D encoder, image and point tokens are generated. Finally, three kinds of tokens are organized and fed into VLM decoder to get final answers corresponding to the CAD file and user's prompt.

alongside textual specification. Furthermore, we build a pipeline (Figure 1) which takes a CAD file and a use instruction as input, automatically samples point clouds and renders multi-view images, and finally outputs responses.

Extensive experiments are conducted to show the effectiveness and strong generalization ability of our proposed model. For generative 3D object classification task on ModelNet40 dataset, our 3B version model surpasses existing methods with 66.49% classification accuracy score, and our 7B model achieves higher to 69.89%. For 3D object captioning task, PointVLM also showcases superior comprehensive performance.

Our contributions can be summarized as follows:

- We introduce a novel pre-training framework based on ULIP to align features from point clouds, images, and text into a unified space.
- We propose **PoinVLM**, a geometric-aware multi-modal architecture which is the first one to our knowledge that bridges the semantic gap between 3D representations and visual-language reasoning.
- We build a pipeline for CAD file pre-processing, point cloud sampling, multi-view image rendering and interaction using instructions.

The remainder of this paper is organized as follows: Section 2 reviews related work, Section 3 details our methodology, Section 4 presents experiments and results, and Section 5 summarizes and discusses future directions.

## 2 RELATED WORK

**Multi-modal large language models.** Multi-modal large language models (MLLMs) have emerged as a transformative paradigm in artificial intelligence, integrating text, images, audio, video or other modality data into unified architectures. These models typically build upon the foundational success of LLMs by incorporating specialized encoders for different modalities, e.g., vision transformer (Dosovitskiy et al., 2020) for images, audio encoder (Radford et al., 2023) for sound, followed by fusion mechanisms such as cross-modal attention and token-level concatenation. Qwen2.5-VL (Bai et al., 2025) introduces dynamic vision resolution handling and absolute time encoding for video processing with an architecture which combines a vision encoder and a multilingual LLM, achieving competitive performance on visual-language tasks. Gemini 2.5 (Comanici et al., 2025)

has the ability to process more than two modalities, including image, video, audio and text. It adopts sparse mixture-of-experts architecture and thinking mechanism, achieving state-of-the-art performance on video understanding and audio generation tasks. In our work, we keep up with the alignment and tuning methods, construct an MLLM capable of understanding 3D object point clouds and images.

**Language models for object point cloud understanding** The integration of language models with 3D point cloud understanding are inspired by works like CLIP (Radford et al., 2021). PointCLIP (Zhang et al., 2022) projects point clouds into multi-view depth maps and aligning them with CLIP's vision-language space. PointCLIP2 (Zhu et al., 2023) extends PointCLIP with an inter-view adapter to aggregate global features and improves few-shot performance. ULIP (Xue et al., 2023) and ULIP2 (Xue et al., 2024) train point cloud encoders to align with CLIP embeddings using triplet data (point clouds, images and text). OpenShape (Liu et al., 2023) combines 2D image features from ResNet with 3D from PointNet++ (Qi et al., 2017), leveraging contrastive learning to align multi-modal features. 3D-LLM (Hong et al., 2023) enables LLMs to interpret 3D scenes by rendering multi-view images and using SAM for object localization but heavily relies on 2D-3D projection pipelines. Point-Bind LLM (Guo et al., 2023) aligns point cloud features with ImageBind's (Girdhar et al., 2023) cros-modal embeddings and uses 2D MLLMs such as ImageBind LLM (Han et al., 2023) for text generation. PointLLM (Xu et al., 2024) directly fuses point cloud tokens with LLMs like LLAMA-3 (Dubey et al., 2024) for 3D object understanding and releases generative 3D object classification and 3D object captioning benchmarks. GreenPLM (Tang et al., 2025) pays more attention to text data and uses less points to reduce reliance on 3D data. However, fusion of visual-language model with point cloud data is insufficiently explored. Aiming at this, our model aligns point cloud tokens along with image and text tokens using an end-to-end structure and training method, enabling free-form interactions while keeping accurate understanding.

## 3 METHODOLOGY

This section firstly introduces pre-training method for point cloud alignment with image and text. We then detail the architecture of PointVLM. Lastly, we introduce our training strategy.

### 3.1 PRE-TRAINING

To better align point cloud features with image and text representations, a pre-training framework (Figure 2) is built. Specifically, a pre-trained vision-language model (CLIP) containing image encoder $f_I(\cdot)$ and text encoder $f_T(\cdot)$ is used to extract image and text features, and a 3D encoder $f_P(\cdot)$ is utilized to get point cloud features. For an CAD triplet input $(I, T, P)_i$, the three en-

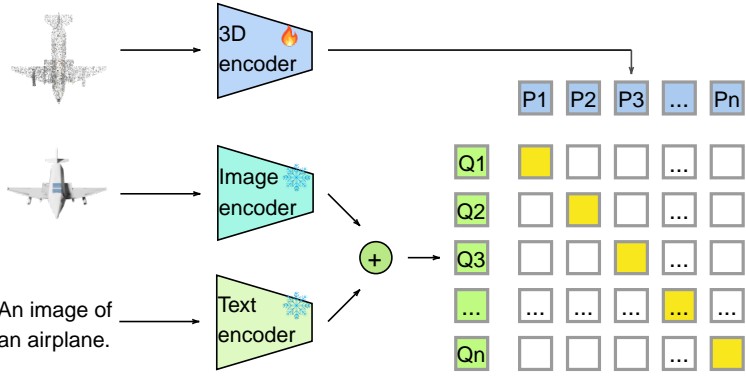

Figure 2: Point cloud, image and text alignment.

coders output corresponding features $X_i = (x_1, x_2, ..., x_n) \in \mathbb{R}^{n \times d}$, $Y_i = (y_1, y_2, ..., y_n) \in \mathbb{R}^{n \times d}$ and $Z_i = (z_1, z_2, ..., z_n) \in \mathbb{R}^{n \times d}$ for image, text and point cloud respectively. During training, only the 3D encoder is trainable, while another two encoders are frozen. To align point cloud features with image and text, we simply add image and text features to get CLIP features $C_i = X_i + Y_i = (x_1 + y_1, x_2 + y_2, ..., x_n + y_n)$. Similar to CLIP, we use contrastive loss to train the model:

$$Loss_{pretrain} = \sum_{(i,j)} (-\frac{1}{2} log \frac{e^{\frac{C_i z_j}{\tau}}}{\sum_k e^{\frac{C_i z_k}{\tau}}} - \frac{1}{2} log \frac{e^{\frac{C_i z_j}{\tau}}}{\sum_k e^{\frac{C_k z_j}{\tau}}}), \tag{1}$$

where $(i, j)$ indicates a positive pair, while $(i, k)$ and $(k, j)$ indicate negative pairs in each training patch. $\tau$ is a learnable temperature as it is in CLIP.

## 3.2 POINTVLM ARCHITECTURE

As shown in Figure 1, our PointVLM is a multi-modal large language model which takes image, point cloud and text as inputs, and generate responses. The model consists of a pre-trained vision encoder $f_V$, a pre-trained 3D encoder $f_P$ which is discussed in Section 3.1, a projector $f_{proj}$ and a pre-trained vision-language model (VLM) decoder $f_{vlm}$.

The pre-trained vision encoder $f_V$ takes as inputs multiple images $\{I\}$, and generates corresponding image tokens $X = (x_1, x_2, ..., x_m) \in \mathbb{R}^{m \times c}$, where $m$ is the number of image tokens and $c$ is the feature dimension. The point encoder $f_P$ takes as input a point cloud $P \in \mathbb{R}^{l' \times d}$, where $l'$ is the number of points and $d$ is the feature dimension of each point. The output of 3D encoder is a vector of point features $Z' = (z'_1, z'_2, ..., z'_l) \in \mathbb{R}^{l \times c'}$, where $l$ is the number of point features and $c'$ is the feature dimension. The projector $f_{proj}$ is a multilayer perceptron (MLP) that maps the point features $Z'$ to point tokens $Z = (z_1, z_2, ..., z_l) \in \mathbb{R}^{l \times c}$, where $c$ is equal to the dimension of image tokens. Additionally, the input texts are tokenized to text tokens $Y = (y_1, y_2, ..., y_n) \in \mathbb{R}^{n \times c}$, where n is the number of text tokens.

All encoded tokens, including image, text and point tokens, are combined into a unified sequence, denoted as $V = (v_1, v_2, ..., v_k) \in \mathbb{R}^{k \times c}$, where $k = m + n + l$. This sequence is fed into the VLM decoder $f_{vlm}$, which can process the mixed-modal tokens, leveraging contextual relationships between image, text and point clouds with self-attention mechanism. The output of VLM decoder is a sequence of predicted tokens $\hat{V} = (\hat{v}_1, \hat{v}_2, ..., \hat{v}_K) \in \mathbb{R}^{K \times c}$, where $K$ is the number of generated tokens util the EOS token or maximum number of truncated tokens. The prediction of the $i$-th token, $\hat{v}_i$, is conditioned on all previous tokens $V_{<i} = (v_1, v_2, ..., v_{i-1})$, expressed mathematically as

$$\hat{v}_i = f_{vlm}(V_{<i}). \tag{2}$$

Finally, to get the prediction for each $\hat{v}_i$, a linear layer followed by a softmax operation is utilized to map it into a probability distribution over the vocabulary. Denote this layer as $f_{head} : \mathbb{R}^c \to \mathbb{R}^W$, where $W$ is the size of the vocabulary, then this process can be expressed as

$$prob_i = \arg \max_{w \in \text{vocab}} f_{head}(\hat{v}_i)[w]. \tag{3}$$

To train the model to predict the next token in a sequence, we utilize the widely used causal language model loss. It computes the loss for autoregressive next-token prediction by aligning input sequences with their shifted labels and applying cross-entropy optimization. This loss function excels in balancing computational efficiency, memory usage and scalability, which makes our training end-to-end and effectively understand point clouds along with images and texts.

## 3.3 TRAINING STRATEGY

Inspired by PointLLM (Xu et al., 2024), we leverage a three-stage training strategy to balance efficiency and performance.

**Pre-training stage.** In the first stage, the purpose is to train the 3D encoder to better extract features from point clouds. During this stage, the 3D encoder is trainable while CLIP image and text encoder are frozen. Point, image and text triplets are fed into this contrastive learning framework to enable 3D encoder's feature extraction ability.

**Feature alignment stage.** In the second stage, we aim at training the MLP projector to map raw point features to semantically meaningful tokens. So, during this stage, only the weights of MLP projector are trainable. We use brief-description instructions with point cloud and text data to train the MLP projector so that it can efficiently adjust to map point features to point tokens. Embedding adjustment for special point tokens ($<$ |point_start| $>$ and $<$ |point_end| $>$) which are used to mark point token boundaries, is also included in this stage.

**Fine-tuning stage.** During stage three, the entire model is frozen to preserve pre-trained knowledge, and low-rank adaptation, LoRA (Hu et al., 2022), is used for each transformer layer. In this stage, complex instructions along with multi-view images and point cloud are fed into the model to enable its ability to understand and respond to complex instructions including point cloud, image and text data. This strategy balances efficiency and performance, making it suitable for deploying large multi-modal models on resource-constrained hardware.

## 4 EXPERIMENTS AND RESULTS

To demonstrate the benefits of our work, we conduct extensive experiments on two downstream 3D tasks: generative 3D object classification and 3D object captioning. In this section we first introduce experiment settings, including our model encoder and decoder backbones, datasets and implementation details. Then we present results of pre-training and downstream tasks, followed by our analyses. Lastly, ablation study and qualitative comparison are shown to demonstrate the effects.

### 4.1 EXPERIMENT SETTINGS

**Backbone networks.** PointBERT (Yu et al., 2022), which is a transformer-based architecture for point cloud feature extraction, is utilized as our 3D encoder backbone. During pre-training stage, CLIP image and text encoder (clip-vit-base-patch32) are used as our image and text backbone. In the feature alignment and fine-tuning stages, we use Qwen2.5-VL vision encoder as image backbone, Qwen2.5-VL decoder as our VLM decoder backbone.

**Datasets.** We conduct pre-training on ShapeNet55 (Chang et al., 2015), which contains around 52.5k samples of 3D objects with 55 category labels. To generate image, text and point cloud triplet, we sample points to construct point cloud from each sample mesh and use a template with its label to generate corresponding text. ModelNet40 (Wu et al., 2015) is a benchmark 3D shape dataset which has 40 categories. We only use the test split which has 2468 samples to conduct zero-shot classification in pre-training evaluation and generative 3D object classification in fine-tuning evaluation. Objaverse (Deitke et al., 2023) is a large-scale 3D dataset containing more than 800k 3D models. By following PointLLM, we use 660k samples with brief descriptions as training data during feature alignment stage, and 70k samples with complex instructions in fine-tuning stage. Additional 200 samples are not seen in training stages, and they are kept as evaluation data for 3D object captioning task. Furthermore, additional samples from Fusion360 (Willis et al., 2021) are used as qualitative comparison.

Table 1: Zero-shot 3D classification comparison on ModelNet40 in pre-training stage.

| Model | Top-1 accuracy (%) |
|---|---|
| PointCLIP | 20.2 |
| PointNet++(ULIP) | 58.4 |
| PointBERT(ULIP) | 60.4 |
| PointVLM(ours) | **71.3** |

Table 2: Generative 3D object classification results on ModelNet40 test split (M40.) and Objaverse 200 samples (Obj.). (I): using instruction-typed prompt "What is this?", (C): using completion-typed prompt "This is an object of ". PCD.: point cloud, SV.: single-view, and MV.: multi-view. For multi-view, we randomly sample 4 views from 12 rendered images.

| Model | Input | M40.(I) | M40.(C) | Obj.(I) | Obj.(C) |
|---|---|---|---|---|---|
| InstructBLIP-7B | SV. Img. | 19.53 | 31.48 | 45.00 | 42.00 |
| InstructBLIP-13B | SV. Img. | 25.97 | 31.40 | 37.00 | 31.50 |
| LLaVA-7B | SV. Img. | 39.75 | 39.67 | 49.50 | 50.50 |
| LLaVA-13B | SV. Img. | 37.12 | 36.06 | 53.00 | 50.50 |
| Point-Bind LLM | PCD. | 51.90 | 39.71 | 6.00 | 4.50 |
| PointLLM-7B | PCD. | 53.44 | 51.82 | 55.00 | 51.00 |
| PointLLM-13B | PCD. | 53.00 | 52.55 | 56.50 | 51.50 |
| GreenPLM | PCD. | 62.60 | 62.68 | 48.00 | 45.00 |
| 3D-LLM | 3D object + MV. Img. | - | - | 49.00 | 41.50 |
| PointVLM-3B(ours) | PCD. + MV. Img. | 65.80 | 66.49 | 54.50 | **57.50** |
| PointVLM-7B(ours) | PCD. + MV. Img. | **69.89** | **68.35** | **57.00** | 57.00 |

**Implementation details.** All our experiments were conducted on a Ubuntu server with 8 Nvidia H20 graphic cards, each with a memory size of 96 GB. For the 3D input, we use number of points $n = 8192$. During pre-training, we use 128 as training batch size and 40 as validation batch size, $10^{-4}$ as the learning rate and trained with 100 epochs. In both feature alignment and fine-tuning stages, cosine learning rate schedule and warm-up strategy with ratio 0.03 are used, the number of epochs is 3. In feature alignment stage, we use 4 as batch size, $2 \times 10^{-3}$ as learning rate. In fine-tuning stage, we use 2 as batch size, $2 \times 10^{-5}$ as learning rate. We use AdamW as optimizer in all three stages. For evaluation metrics, in pre-training stage, top-1 accuracy is used. For generative 3D object classification and 3D object captioning tasks, large language model (Gemini 2.5 Flash) is used to evaluate results. Details on how Gemini is used can be found in the appendix.

## 4.2 RESULTS AND ANALYSES

**Pre-training results.** We present the zero-shot 3D classification results on ModelNet40 in Table 1. As can be seen, our method outperforms existing models with top-1 accuracy of about 71.3%, which outperforms ULIP by around 10.9%. It indicates that by integrating image and text features into one vector, the performance improves. While during evaluation in ULIP method, image features are not used, which makes it hard because during training, point features are aligned to both image and text features.

Table 3: 3D object captioning results on Objaverse 200 samples. We report LLM-score evaluated by Gemini, S-BERT which refers to sentence BERT score, and SimCSE score.

| Model | LLM-Score | S-BERT | SimCSE |
|---|---|---|---|
| LLaVA-7B | 46.71 | 45.61 | 47.10 |
| LLaVA-13B | 38.28 | 46.37 | 45.90 |
| 3D-LLM | 33.42 | 44.48 | 43.68 |
| PointLLM-7B | 44.85 | 47.47 | 48.55 |
| PointLLM-13B | 48.15 | **47.91** | 49.12 |
| PointVLM-3B(ours) | 63.08 | 40.30 | 51.52 |
| PointVLM-7B(ours) | **68.59** | 42.18 | **52.26** |

**Generative 3D object classification results.** What makes this task different from zero-shot 3D classification and more challenging is that category names are unknown during inference. On generative 3D object classification task, we compare various models as shown in Table 2. To stay consistent with PointLLM, we use two kinds of prompts: the instruction-typed prompt "What is this?" and the completion-typed prompt "This is an object of ". As can be seen, both on ModelNet40 and Objaverse dataset, our method outperforms existing models. For models relying on single-view images, their performance is notably constrained. Even the larger-scale LLaVA-13B only reaches 37.12 and 53.00, which shows that single-view visual cues struggle to encode the comprehensive 3D geometry and semantic information needed for this task. Among point cloud models, although PointLLM-7B/13B and GreenPLM perform better, their results still fall short of our approach. On contrast, our PointVLM models leverage the synergy of point cloud and multi-view images. On Model-Net40, they achieve remarkable scores: PointVLM-3B attains 65.80 and 66.49, while PointVLM-7B reaches 69.89 and 68.35, all of which are the highest in their respective categories. On Objaverse, our models also lead in most scenarios: PointVLM-3B scores 54.50 and 57.50, and PointVLM-7B achieves 57.00. This dominance stems from one key factor: the multi-modal input compensates for the limitations of single-view or single-modality data, enabling richer feature extraction of 3D objects. In summary, the integration of multi-modal inputs and our innovative approach empowers PointVLM to set new benchmarks in generative 3D object classification.

**3D object captioning.** On 3D object captioning task, we evaluate our model with the same 200 samples from Objaverse across three metrics of LLM-score (evaluated by GPT/Gemini), sentence BERT score, and SimCSE. Notably, the PointVLM series proposed in this study stands out. Among all compared models, PointVLM-3B achieves an LLM-score of 63.08, and PointVLM-7B further improves to 68.59, significantly outperforming other baselines, and even the PointLLM series (PointLLM-7B: 44.85; PointLLM-13B: 48.15) in terms of LLM-score. While in sentence BERT score, although PointVLM models do not claim the top spot (PointLLM-13B reaches 47.91), they remain competitive with scores of 40.30 (PointVLM-3B) and 42.18 (PointVLM-7B). In SimCSE score, PointVLM-7B hits 52.26, ranking among the leading results. Overall, the PointVLM series showcases superior comprehensive performance in 3D object captioning, especially excelling in the LLM-score metric, which verifies the effectiveness of our proposed approach.

### 4.3 Ablation study and qualitative comparison

**Ablation study.** We conducted a study focusing on the number of image views during inference using PointVLM-3B. The results show distinct trends, increasing the number of image views from 1 to 2 brings a remarkable performance improvement (from 46.50 to 55.00 and from 50.50 to 54.50). Further increasing to 4 images leads to a slight drop to 54.50 for instruction-typed prompt, but the performance keeps rising to 57.50 with 4 images. These results indicate that the advantage of additional image views helps models understand 3D geometry better.

**Qualitative comparison.** Table 5 and Table 6 show qualitative results compared with PointLLM. In Table 5, for Sample 1 from ModelNet40 test split, PointLLM erroneously describes it as a minimalist grey bowl, while PointVLM accurately identifies it as a cartoon-styled bathtub. For Sample 2 from Objaverse 200 samples, where the ground truth involves a black cat with yellow eyes chasing a green ball, PointLLM fails to capture this accurately and describes a cartoon pig instead. PointVLM, however, correctly describes the cat and the ball. Furthermore, in Table 6, for Sample 3 from Objaverse 200 samples, PointLLM describes the 3D model as a toy balloon. In contrast, PointVLM depicts it

Table 4: Ablation study. We show the results under different number of image views during inference. Obj. is Objaverse 200 samples. (I) indicates using instruction-typed prompt "What is this?" and (C) means using completion-typed prompt "This is an object of ".

| Model | Input | Obj.(I) | Obj.(C) |
|---|---|---|---|
| PointVLM-3B | PCD. + 1 Img. | 46.50 | 50.50 |
| PointVLM-3B | PCD. + 2 Img. | **55.00** | 54.50 |
| PointVLM-3B | PCD. + 4 Img. | 54.50 | **57.50** |

Table 5: Qualitative comparison: instruction-typed and completion-typed. Sample 1 is from ModelNet40 test split, and sample 2 is from Objaverse 200 samples. Our PointVLM generates more accurate results than PointLLM.

| | | |
|---|---|---|
| Sample 1, 2 | 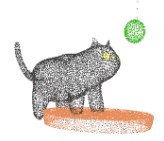 | 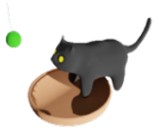 |
| Ground truth | Bathtub | Figure of a black cat with yellow eyes following a small green ball that is suspended by a very thin white rope. The cat stands on two legs on a round light brown surface. |
| Prompt | What is this? | This is an object of |
| PointLLM-13B | The 3D model presents an unornamented, minimalist grey **bowl**. It's simple yet functional design includes a flat, smooth top surface for placing items, with a slightly wider base for stability. The color and lack of embellishment make it versatile, suitable for multiple uses in different contexts - from a simple household bowl to a decorative pot or even a pedestal for plants. It is empty and is waiting for your creative decor or practical application. | a whimsical and charming cartoon-style **pig**. The pig is characterized by a vivid orange body and a distinctive pink face, creating a playful contrast. Highlighting its captivating features are luminous yellow eyes which add an extra layer of brightness. This 3D model, while being visually appealing, can also be used in various digital mediums such as animation, video games or even virtual reality. |
| PointVLM-3B | This is a 3D model of a **bathtub**, designed in a cartoon style. The bathtub is white and has a simple, clean design. It features two handles for easy access to the water, which are located on either side of the tub. This model could be used in animations, video games, or other digital media where realistic objects need to be represented in a simplified manner. | a **black cat** playing with a **green ball**. The cat is standing on a **brown, circular base** that resembles a platform or a cushion. The scene suggests a playful and interactive moment between the cat and its toy. |

as a cartoon-style submarine with a yellow body, blue eyes, and a red propeller. For Sample 4 from Fusion360 dataset, PointLLM captions it as a door handle, while PointVLM captions it as a modern telephone. These examples clearly demonstrate that PointVLM captures topological relationships and outperforms PointLLM, even with less model weights.

## 5 CONCLUSION AND FUTURE WORKS

In this work, we present PointVLM, a novel multi-modal vision-language model that bridges 3D point cloud processing with vision and natural language understanding to enable precise CAD model interpretation. Experimental results validate that PointVLM outperforms existing methods. The

Table 6: Qualitative comparison: captioning. Sample 3 is from Objaverse 200 samples, and sample 4 is from Fusion360 dataset. The prompt for them is "Caption this 3D model in detail.". Our PointVLM outperforms PointLLM.

| Sample 3, 4 | 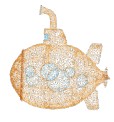 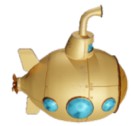 | 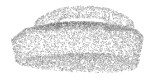 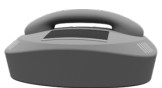 |
|---|---|---|
| PointLLM-13B | The 3D model is of a vibrant, cheerful toy **balloon** in a striking shade of yellow, which would instantly attract young children. The balloon is embellished with multiple tiny, blue dots evenly scattered all over its body, creating an eye-catching visual effect. It is further embellished with an unusually lengthy, tubular shape that starts at the bottom and stretches all the way up to the pointed apex of the balloon, possibly serving as an additional decorative feature or a handle for lifting. Its playful aesthetic indicates it's primarily meant for use in children's games or as a cheerful display. | The model is a simplistic, minimalistic representation of a door **handle**, primarily in a grey color. As a 3D object, it can be distinguished from the traditional 2D design, adding depth and texture to the handle. The handle is designed with smooth surfaces, giving it a sleek and modern appearance. The 3D model is implied to be functional, similar to its 2D counterpart, allowing doors to be opened and closed. It's likely to be used in a 3D environment like a video game, an architectural model, or a movie. |
| PointVLM-3B | This is a 3D model of a cartoon-style **submarine**, characterized by its vibrant yellow color and playful design. The submarine features two large blue eyes on the sides, giving it a friendly and engaging appearance. It also has a distinctive red propeller at the back, adding to its whimsical look. This model could be used in animations, video games, or as a decorative piece for children's rooms. | The 3D model represents a modern, sleek, and compact **telephone** with a distinctive design. The phone is primarily grey in color, giving it a professional and sophisticated look. It features a large, rectangular screen at the front, which likely serves as the display for calls or messages. The phone also has a built-in speakerphone, indicated by the presence of a small, circular hole on top. This design suggests that the phone is intended for use in both home and office environments, offering convenience and functionality to users. |

integration of 3D spatial reasoning with vision-language models enables robust performance across CAD understanding scenarios.

To further advance CAD model interpretation and multi-modal AI systems, following directions could be explored in the future: 1) Inspired by recent advancements in reasoning-aware VLMs trained with chain-of-thought dataset, we could develop a reasoning framework that decomposes complex CAD analysis tasks into explainable cognitive steps. 2) Reinforcement learning could be utilized in the future to improve generalization. 3) Exploring generative CAD capabilities for automated and controllable 3D model editing could be another future direction.

ACKNOWLEDGMENTS

Masked due to double-blind review.

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

# A APPENDIX

## A.1 TEMPLATES OF PRE-TRAINING

During pre-training stage on ShapeNet55, to construct image, text and point cloud triplets, we use object labels and randomly select one template from Table 7 to generate corresponding text. The words split by "/" in every template are also random selected when generating samples.

Table 7: Templates of text used in ShapeNet55 dataset to construct image, text and point cloud triplets. {} will be replaced with corresponding labels when sampling.

| | |
|---|---|
| A point cloud model of {}. | There is a/the {} in the scene. |
| A photo/model of a/the/one {} in the scene. | A photo/model of a/my/the/one/many {}. |
| A good photo/model of a/the {}. | A bad photo/model of a/the {}. |
| A photo/model of a/the nice {}. | A photo/model of a/the cool {}. |
| A photo/model of a/the weird {}. | A photo/model of a/the small {}. |
| A photo/model of a/the large {}. | A photo/model of a/the clean {}. |
| A photo/model of a/the dirty {}. | A bright photo/model of a/the {}. |
| A dark photo/model of a/the {}. | A photo/model of a/the hard to see {}. |
| A low resolution photo/model of a/the {}. | A cropped photo/model of a/the {}. |
| A close-up photo/model of a/the {}. | A jpeg corrupted photo/model of a/the {}. |
| A blurry photo/model of a/the {}. | A pixelated photo/model of a/the {}. |
| A black and white photo/model of a/the {}. | A/The plastic {}. |
| A/The toy {}. | A/The plushie {}. |
| A/The cartoon {}. | An/The embroidered {}. |
| A painting/modeling of a/the {}. | |

## A.2 LLM EVALUATION PROMPTS AND USE OF LLMS

Inspired by PointLLM, we use Gemini 2.5 Flash as our LLM evaluator to help use evaluate our results. Table 8 shows the prompt that we used for close-set zero-shot classification task on ModelNet40. Gemini is asked to directly give an answer containing category index, category name and brief reason according to model output. Table 9 shows the prompt for open vocabulary classification on Objaverse 200 samples. Gemini is given two sentences to determine if they are referring to the same general object or concept, and answer True or False followed by a brief reason. Table 10 shows the prompt for evaluating captioning task. Gemini is asked to score a model-generated caption according to human caption, by counting mentioned aspects.

**Use of LLMs.** It is important to note that, in this work, LLMs, specifically, Gemini 2.5 Flash, was only used to help evaluate experimental results. The core methodology did not involve the use of LLM-generated content.

## A.3 MORE QUALITATIVE RESULTS

We provide more qualitative results from different datasets of PointVLM 3B. All samples used were unseen by our models during training. Table 11 shows two samples from Objaverse 200 samples. Sample 5 is a 3D model of a forklift, but PointLLM captions it as a truck while PointVLM successfully recognize it as a forklift. Sample 6 is a carpet, and PointLLM misidentifies it as a keyboard. Table 12 shows another two samples from Fusion360 dataset. The captions of PointVLM are more accurate than PointLLM for sample 7 and 8 (pliers and pipe wrench). It is worth noting that PointVLM uses 3B parameters only, while PointLLM uses 13B parameters. These samples highlight PointVLM's generalization ability and efficiency.

Table 8: Prompt of Gemini in close-set zero-shot classification evaluation on ModelNet40 test split.

| | |
|---|---|
| Prompt | Given the following free-form description of a 3D object, please determine the most probable class index from the following 40 available categories, even if the description doesn't clearly refer to any one of them. Make your best-educated guess based on the information provided. If the description already contains a valid index, then the index should be selected. If it contains more than one valid index, then randomly select one index (specify your reason). If there is no valid index and it cannot be inferred from the information, return "-1#NA#Cannot infer". |
| | Categories:
0: airplane, 1: bathtub, 2: bed, 3: bench, 4: bookshelf, 5: bottle, 6: bowl, 7: car, 8: chair, 9: cone, 10: cup, 11: curtain, 12: desk, 13: door, 14: dresser, 15: flower_pot, 16: glass_box, 17: guitar, 18: keyboard, 19: lamp, 20: laptop, 21: mantel, 22: monitor, 23: night_stand, 24: person, 25: piano, 26: plant, 27: radio, 28: range_hood, 29: sink, 30: sofa, 31: stairs, 32: stool, 33: table, 34: tent, 35: toilet, 36: tv_stand, 37: vase, 38: wardrobe, 39: xbox
Examples: |
| | Input: This is a 3D object model of a cartoon white truck.
Output: 7#car#Closest match to "car"in categories. |
| | Input: A green leaf in a flower pot.
Output: 26#plant#The primary subject "leaf"directly indicates a plant. |
| | Input: It's difficult to determine the exact type of this object due to insufficient details. But it seems to be like a piece of furniture.
Output: 33#table#Randomly select one kind of furniture from the list. |
| | Input: I cannot determine the specific type of the object without additional information or context.
Output: -1#NA#Cannot infer. |
| | Now analyze the following:
Input: {model_output}
Output: |
| Example 1 | Input: This is a model of an airplane, designed in a cartoon style. It's predominantly white and has a playful, simplified design that makes it suitable for children's entertainment or educational purposes. The airplane features two wings, a tail, and a cockpit area, all typical components of a real aircraft. Its cartoonish appearance suggests it might be used in animations, video games, or as a teaching tool to explain basic concepts about aviation.
Output: **0#airplane**#The description explicitly states "This is a model of an airplane". |
| Example 2 | Input: This is a 3D model of a white gaming console, which appears to be a Wii console based on its design and features. The console has a distinctive rectangular shape with a control pad attached to it. This type of console was popular for its motion-sensing capabilities, allowing players to interact with games using physical movements.
Output: **39#xbox**#The description refers to a "gaming console" and specifically mentions a "Wii console" which is a type of gaming console. "xbox" is the closest category for a gaming console. |
| Example 3 | Input: This is a 3D model of a book, which is open to reveal two blank pages. The book appears to be made of paper and has a clean, white cover. It's a simple, minimalist design that could be used in various digital contexts such as animations, games, or graphic designs.
Output: **-1#NA**#Cannot infer. |

Table 9: Prompt of Gemini in open vocabulary classification evaluation on Objaverse 200 samples.

| | |
|---|---|
| Prompt | Analyze two sentences and determine if they're referring to the same general object or concept, focusing on the type of object, not attributes such as color, size, or shape. Respond with "T"if they refer to the same thing and "F"if not. Also, provide a brief rationale (no more than 20 words) for your judgment.

Example:
Input: 1. Spiral staircase that goes from a ground floor. 2. This is a 3D model of wooden stairs in light brown
Output: T#Both refer to a staircase.

Now, analyze the following:
Input: 1. {ground_truth} 2. {model_output}
Output: |
| Example 1 | Input: 1. A cartoon black carpet in 3d. 2. This 3D model is a vibrant and colourful representation of a cartoon-like keyboard. It is adorned with buttons of varied colours that give it a lively and playful aesthetic. The model appears to be designed for children, emphasizing on the fun aspect of learning or using a keyboard. Although it doesn't showcase any functionalities, it can be assumed that it's used for typing or gaming in a digital environment. Its child-friendly look can be instrumental in engaging younger audiences in educational or entertainment scenarios.
Output: **F**#One describes a carpet, the other a keyboard. |
| Example 2 | Input: 1. A white container like car with six black tractor legs and yellow sides. 2. The 3D model represents a brown cartoon-style truck. The truck features large black tires with yellow rims that provide a playful contrast to the overall brown body. It also has a charming little yellow headlight in the front, which adds more personality to its cartoonish design. The truck could be used in animations, video games, or children's book illustrations.
Output: **T**#Both describe a truck. |

Table 10: Prompt of Gemini in 3D object captioning evaluation on Objaverse 200 samples.

| | |
|---|---|
| Prompt | Evaluate a model-generated caption against a human-generated caption (ground truth) for a 3D model. Identify the aspects mentioned in the human caption and calculate the percentage of these aspects correctly mentioned or partially matched in the model caption. Score from 0 to 100, where each aspect contributes equally to the score. Consider similar concepts for a partial score.

Provide your score (0-100) and a short justification (less than 15 words) in the format of "score#reason"

Example:
Human: A white brown skeleton
Model: This is a 3D model of a small, cartoon-like robot. It has a spherical body and is covered in a layer of white dust.
Output: 50#mention white; skeleton and robot have similar appearence.

Now score the following:
Human: {ground_truth}
Model: {model_output}
Output: |
| Example 1 | Human: A blue cartoon.
Model: This is a 3D model of a cartoon-style pony, characterized by its vibrant blue color and striking white wings. The pony's design is whimsical and playful, with a crown adorning its head, adding to its regal appearance. Its wings, which are large and spread wide, suggest that it might be capable of flight or at least soaring through the air. This model could be used in animations, video games, or other digital media for children's entertainment purposes.
Output: **100**#Correctly mentions both 'blue' and 'cartoon'. |
| Example 2 | Human: Digital image for the tr-909 drum machine by roland brand with 30 dials and 8 scuare yellow buttons , 6 little scuare white buttons and 18 white buttons 16 buttons with leds light and a red scuare power button.
Model: The 3D model represents a rectangular, black-colored device with a sleek design. It features a series of buttons and switches arranged in a grid pattern on its surface, suggesting it's a control panel or a switchboard. The buttons and switches appear to be functional, indicating that the device could be used for controlling various systems or devices. The overall appearance suggests it might be used in industrial or commercial settings where precise control over multiple functions is required.
Output: **44**#Partially matches device type & components, but misses brand, colors, counts, and features. |
| Example 3 | Human: A flat grey object with a key hole gap in the top and having two flat regions in the back.
Model: The 3D model is of a black, cartoon-style spaceship with a distinctive 'X' shape. It has two large, circular windows on the front and a smaller one at the back. The design suggests it's from a science fiction or fantasy genre, possibly used for entertainment purposes such as video games, animations, or children's books.
Output: **0**#No matching aspects; descriptions are completely different. |

Table 11: More qualitative comparison from Objaverse 200 samples. The prompt for them is "Caption this 3D model in detail.".

| Sample 5, 6 | 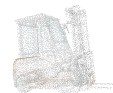 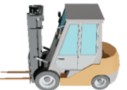 | 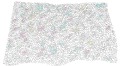 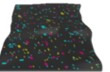 |
|---|---|---|
| PointLLM-13B | The model represents a detailed 3D view of a cargo **truck**, primarily colored in a distinctive orange hue. The truck is equipped with a vibrant blue trailer that has an extensive, rectangular cargo area. The truck driver's cabin is situated at the front of the trailer. Cargo trucks like this are commonly used for transporting goods over long distances. With its vibrant color scheme, this model could be useful in providing educational material to children or could be used as an attractive piece of decor. | This 3D model is a vibrant and colourful representation of a cartoon-like **keyboard**. It is adorned with buttons of varied colours that give it a lively and playful aesthetic. The model appears to be designed for children, emphasizing on the fun aspect of learning or using a keyboard. Although it doesn't showcase any functionalities, it can be assumed that it's used for typing or gaming in a digital environment. Its child-friendly look can be instrumental in engaging younger audiences in educational or entertainment scenarios. |
| PointVLM-3B | This is a 3D model of a **forklift**, characterized by its robust and sturdy build. The forklift features a grey cabin with two transparent windows, providing visibility to the operator. It has a brown base, which is typical for forklifts designed for heavy-duty tasks. The forklift's design suggests it's used in industrial settings for lifting and moving heavy objects. | The 3D model is a rectangular piece of **fabric**, predominantly black in color. It features a pattern of multi-colored dots scattered across its surface. The dots come in various colors including red, blue, green, and yellow, adding a vibrant contrast to the black background. This fabric could be used for a variety of purposes such as clothing, home decor, or even as a base material for crafting projects. |

Table 12: More qualitative comparison from Fusion360 dataset. The prompt for them is "Caption this 3D model in detail.".

| Sample 7, 8 | 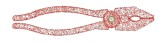 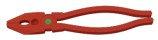 | 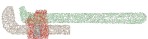 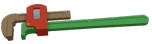 |
|---|---|---|
| PointLLM-13B | A vivid and lively cartoon-style 3D model that takes on the appearance of a pair of **scissors**. It features the blades and handles, traditionally colored in grey and red respectively, portraying the familiar shape and functionality of real-life scissors. It is set against a plain gray background, making the object stand out. Due to its cartoonish design, this model could be used in animations or children's educational content, emphasizing its role as a cutting tool in everyday life. | This is a three-dimensional model of a cartoon-style, green-colored object that resembles a tool. The model is quite simple and abstract, giving it a playful and whimsical look typical of cartoons. The specific type of tool it represents is **not clear** due to its vague design, but it could be any physical object used for laborious tasks in daily life, such as a hammer, wrench, or screwdriver. |
| PointVLM-3B | The 3D model represents a pair of **pliers**, characterized by its long, slender handles and sharp, curved jaws. The handles are designed for comfortable gripping, while the jaws are equipped with teeth for gripping and cutting. This tool is commonly used in various tasks such as assembling or disassembling objects, cutting wires, or performing other manual work. Its design suggests it's made from durable materials like metal, ensuring longevity and strength. | The 3D model depicts a cartoon-style **pipe wrench**, characterized by its exaggerated proportions and vibrant colors. The wrench is primarily green, with a red handle that features a distinctive square-shaped grip area. The design suggests it's meant for children's play or educational purposes, possibly to teach them about tools in a fun and engaging way. |

