# OpenReview forum: "PointVLM: Multi-Modal Vision-Language Model for CAD Model Understanding via Point Cloud Integration"
_ICLR.cc/2026/Conference — Submitted to ICLR 2026_

### Official Review · Reviewer_rdwm · 2025-10-23

**Soundness:** 2
**Presentation:** 3
**Contribution:** 1
**Rating:** 2
**Confidence:** 3

**Summary:**

This paper proposes a multi-modal vision-language model designed to interpret complex CAD models by integrating 3D point cloud, image, and text modalities. Built on the Qwen2.5-VL architecture, PointVLM employs a learnable projector to fuse geometric and semantic features for context-aware understanding. The training pipeline consists of three stages: pre-training the 3D encoder through contrastive learning, aligning point features via an MLP projector, and fine-tuning the model using LoRA for efficient multi-modal adaptation.

**Strengths:**

1. Paper is clearly stated and easy to follow.

**Weaknesses:**

1. The term “CAD file” used in the title and throughout the paper is somewhat confusing or potentially inappropriate. Typically, Computer-Aided Design (CAD) refers to any digital geometric representation created using CAD software such as SolidWorks, AutoCAD, Fusion 360, CATIA, NX, or Rhino. It seems that "CAD file" mentioned here refers to general object mesh as illustrated in Figure 1 and mentioned in Section4.1.

2. The proposed “pre-training” in Section 3.1 of this paper appears to be very similar to the pretraining approach used in ULIP:

- ULIP: Learning a Unified Representation of Language, Images, and Point Clouds for 3D Understanding, Xue et al. (https://arxiv.org/pdf/2212.05171)

3. Sections 3.2 and 3.3 appear to describe a standard multimodal model training approach, similar to PointLLM in concatenating different modalities, and lack clear novelty.

- PointLLM: Empowering Large Language Models to Understand Point Clouds, Xu et al. (https://arxiv.org/abs/2308.16911)

4. The experimental setup in Table 1, described in Lines 304–309, is unclear. What is the input to PointVLM? The reviewer concerns that excluding image features during inference with ULIP may lead to an unfair comparison.

5. The comparison in Table 2 is not entirely reasonable, as it involves methods or models using different types of inputs.

6. Table 3 does not clearly demonstrate that the proposed method outperforms all baselines.

7. In Section 4.3, testing with different numbers of images is inappropriate to label as an “Ablation” study because it does not isolate or remove specific components of the model to measure their individual contributions; rather, it examines the model’s sensitivity to varying input conditions.

8. Typo in L88-L89: We propose **PoinVLM**, a geometric-aware multi-modal architecture which is the first one to our knowledge that bridges the semantic gap between 3D representations and visual-language reasoning.

**Questions:**

See Weaknesses.

---

### Official Review · Reviewer_tbPG · 2025-10-28

**Soundness:** 2
**Presentation:** 3
**Contribution:** 2
**Rating:** 2
**Confidence:** 5

**Summary:**

This paper presents  PointVLM, a multi-modal vision-language model that bridges 3D point cloud processing with vision and natural
language understanding. The proposed method is evaluated by 3D object classification and 3D object captioning tasks.

**Strengths:**

The paper is well-written, and the explanations are clear and easy to follow.

**Weaknesses:**

1. The contributions are significantly over-claimed:

* The first contribution, the pre-training framework, is essentially the same as ULIP pre-training. No new or novel components have been introduced.

* The second contribution, claiming to be the first to bridge the semantic gap between 3D representations and visual–language reasoning, is not true. Prior works such as ShapeLLM[1] have already addressed this problem and achieved remarkable performance.

* The third contribution, the pipeline for processing CAD files, has already been implemented by ULIP, which also open-sourced all the processed data.

2. The experiments are limited and unfair:

* For the zero-shot 3D classification task, the paper only compares against ULIP. However, many follow-up works (e.g., OpenShape[2], TAMM[3], Uni3D[4], etc.) have significantly improved upon ULIP but are not included in the comparisons.

* The paper does not report any results on the Objaverse-LVIS dataset for zero-shot 3D classification, which is a more challenging and long-tailed benchmark.


[1] ShapeLLM: Universal 3D Object Understanding for Embodied Interaction

[2] OpenShape: Scaling Up 3D Shape Representation Towards Open-World Understanding

[3] TAMM: Triadapter multi-modal learning for 3d shape understanding

[4] Uni3D: Exploring Unified 3D Representation at Scale

**Questions:**

See above weakness.

---

### Official Review · Reviewer_TriF · 2025-10-31

**Soundness:** 2
**Presentation:** 3
**Contribution:** 2
**Rating:** 4
**Confidence:** 3

**Summary:**

This paper introduces PointVLM, a multimodal large language model that understands 3D objects by fusing 3D point clouds, multi-view 2D images, and text. The model incorporates PointBERT as its 3D encoder and is based on Qwen2.5-VL. The authors propose a three-stage training strategy to align multimodal features. Experimental results show that the model achieves state-of-the-art performance in generative 3D object classification on ModelNet40 and 3D object captioning tasks on Objaverse.

**Strengths:**

1.	This paper tackles the critical challenge of integrating 3D representations into multimodal large models and highlights the limitations of relying solely on 2D images. Concentrating on applications involving complex CAD models further underscores the research’s significance.
2.	The model achieved state-of-the-art performance on widely recognized benchmarks. For example, in the ModelNet40 generative 3D object classification task, PointVLM-7B achieved an accuracy of 69.89%, significantly surpassing existing methods. This demonstrates the model’s exceptional engineering and implementation.

**Weaknesses:**

1.	This architecture is essentially a direct integration of existing mature components (PointBERT, Qwen2.5-VL) and lacks methodological innovation. The model's strong performance likely arises mainly from its robust VLM foundation. However, the paper lacks sufficient ablation studies and notably fails to provide a baseline using only multi-view images for comparison. Consequently, it is impossible to assess the true contribution of the point cloud modality or to validate the architectural design's effectiveness.
2.	The paper claims to address the challenge of understanding “intricate multi-object systems with hierarchical assembly relationships, interpreting complex CAD models”. However, all quantitative experiments were conducted solely on the ModelNet40 and Objaverse datasets, which contain only single, simple objects. Consequently, the paper's core claim of understanding complex CAD models is unsupported by experimental evidence.
3.	The paper primarily uses an LLM-Score derived from a closed-source API as its evaluation metric. This approach lacks reproducibility and may introduce bias. In the task 3D object captioning, PointVLM's LLM-Score significantly outperforms the baseline, whereas its S-BERT score is lower. This discrepancy likely arises because the LLM-Score favors the foundation model's fluent linguistic style over content accuracy, which undermines the validity of the evaluation results.

**Questions:**

1. Beyond using a more powerful VLM foundation, what specific methodological innovations does PointVLM offer compared to prior works?
2. How do you demonstrate that your model can handle “hierarchical assembly relationships” when all experiments are performed only on simple, single objects?
3. How do you account for the significant divergence between LLM-Score and S-BERT scores? How can we be confident that a higher LLM-Score reflects improved content accuracy rather than a preference for linguistic style?
4. To accurately evaluate the contribution of the 3D modality, could you present a key ablation study comparing your model with a baseline that uses only multi-view images on the same VLM base?

---

### Official Review · Reviewer_XG36 · 2025-11-01

**Soundness:** 2
**Presentation:** 2
**Contribution:** 1
**Rating:** 0
**Confidence:** 5

**Summary:**

This work introduces a point-based multimodal large language model (MLLM) named PointVLM, which integrates point clouds, images, and language. The model first pre-trains its point cloud encoder using the ULIP-2 strategy, and then leverages point cloud datasets such as Objaverse to align point clouds and images with large language models, following a similar approach to the previous PointLLM framework. Experimental results demonstrate that PointVLM achieves superior performance compared to prior baselines.

**Strengths:**

1. The writing is clear and easy to follow.
2. The experimental setup is fair and well-designed.

**Weaknesses:**

1. For most parts, this work largely follows the previous study PointLLM, including the pre-training of the point encoder using ULIP, the alignment strategy between point clouds and LLMs, the datasets used for training, and the evaluation settings. As a result, the novelty of this work is extremely limited.

- The paper claims to introduce a novel pre-training framework based on ULIP. However, I do not observe any real novelty here—it appears almost identical to the original ULIP and ULIP-2 papers. Moreover, such pre-training has already been adopted by previous work, including PointLLM.

- The only difference I can identify between PointVLM and PointLLM is that PointVLM uses Qwen-VL as the backbone and supports image inputs in addition to point clouds. However, the integration of images and point clouds has already been explored in several prior works, such as [LEO](https://arxiv.org/abs/2311.12871) and [ChatScene](https://proceedings.neurips.cc/paper_files/paper/2024/hash/cebbd24f1e50bcb63d015611fe0fe767-Abstract-Conference.html).

- The third claimed contribution is a pipeline for CAD processing to generate point cloud–image pairs and language triplets. However, the paper does not describe this proposed pipeline. Furthermore, the pre-trained dataset used is ShapeNet55, for which the previous ULIP work already provides the required triplets. Therefore, I strongly doubt the validity of this claimed contribution.

Overall, this work fails to properly discuss and cite related prior research, and seems to present existing efforts as its own contributions. I would recommend strong rejection for this submission.

It should be noted that the writing of 3.2 PointVLM Architecture is quite similar to that of the 3.2 Model Architecture of the PointLLM paper (https://arxiv.org/pdf/2308.16911).

**Questions:**

/

**Details Of Ethics Concerns:**

The writing of 3.2 PointVLM Architecture is quite similar to that of the 3.2 Model Architecture of the PointLLM paper (https://arxiv.org/pdf/2308.16911).

---

### Meta-Review · Area_Chair_ptFo · 2025-12-10

**Summary:**

This paper presents a multimodal vision–language model for interpreting CAD models by fusing 3D point clouds, images, and texts. All reviewers assigned negative scores, and the authors did not provide a rebuttal.

**Reviewer Concerns:**

Reviewers note that the work closely follows existing methods such as PointLLM and ULIP, offering only minor modifications and thus limited novelty. Additional concerns include over-claimed contributions, an unclear data pipeline, and insufficient evaluation.

**Reviewer Scores:**

Reviewers XG36, TriF, tbPG, and rdwm gave initial scores of 0, 4, 2, and 2, respectively; due to the absence of an author response, all reviewers are expected to retain their original scores.

---

### Decision · Program_Chairs · 2026-01-26

Reject